# On Warm-Starting Neural Network Training

**Jordan T. Ash**
Microsoft Research NYC
ash.jordan@microsoft.com

**Ryan P. Adams**
Princeton University
rpa@princeton.edu

## Abstract

In many real-world deployments of machine learning systems, data arrive piece-meal. These learning scenarios may be passive, where data arrive incrementally due to structural properties of the problem (e.g., daily financial data) or active, where samples are selected according to a measure of their quality (e.g., experimental design). In both of these cases, we are building a sequence of models that incorporate an increasing amount of data. We would like each of these models in the sequence to be performant and take advantage of all the data that are available to that point. Conventional intuition suggests that when solving a sequence of related optimization problems of this form, it should be possible to initialize using the solution of the previous iterate—to "warm start" the optimization rather than initialize from scratch—and see reductions in wall-clock time. However, in practice this warm-starting seems to yield poorer generalization performance than models that have fresh random initializations, even though the final training losses are similar. While it appears that some hyperparameter settings allow a practitioner to close this generalization gap, they seem to only do so in regimes that damage the wall-clock gains of the warm start. Nevertheless, it is highly desirable to be able to warm-start neural network training, as it would dramatically reduce the resource usage associated with the construction of performant deep learning systems. In this work, we take a closer look at this empirical phenomenon and try to understand when and how it occurs. We also provide a surprisingly simple trick that overcomes this pathology in several important situations, and present experiments that elucidate some of its properties.

## 1   Introduction

Although machine learning research generally assumes a fixed set of training data, real life is more complicated. One common scenario is where a production ML system must be constantly updated with new data. This situation occurs in finance, online advertising, recommendation systems, fraud detection, and many other domains where machine learning systems are used for prediction and decision making in the real world [1–3]. When new data arrive, the model needs to be updated so that it can be as accurate as possible and account for any domain shift that is occurring.

As a concrete example, consider a large-scale social media website, to which users are constantly uploading images and text. The company requires up-to-the-minute predictive models in order to recommend content, filter out inappropriate media, and select advertisements. There might be millions of new data arriving every day, which need to be rapidly incorporated into production ML pipelines.

It is natural in this scenario to imagine maintaining a single model that is updated with the latest data at regular cadence. Every day, for example, new training might be performed on the model with the updated, larger dataset. Ideally, this new training procedure is initialized from the parameters of yesterday's model, i.e., it is "warm-started" from those parameters rather than given a fresh initialization. Such an initialization makes intuitive sense: the data used yesterday are mostly the same as the data today, and it seems wasteful to throw away all previous computation. For convex optimization problems, warm starting is widely used and highly successful (e.g., [1]), and the theoretical properties of online learning are well understood.

However, warm-starting seems to hurt generalization in deep neural networks. This is particularly troubling because warm-starting *does not* damage training accuracy.

Figure 1 illustrates this phenomenon. Three 18-layer ResNets have been trained on the CIFAR-10 natural image classification task to create these figures. One was trained on 100% of the data, one was trained on 50% of the data, and a third warm-started model was trained on 100% of the data but initialized from the parameters found from the 50% trained model. All three achieve the upper bound on training accuracy. However, the warm-started network performs worse on test samples than

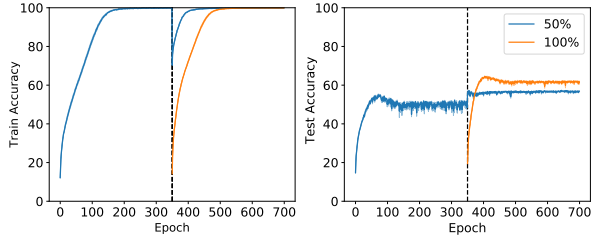

Figure 1: A comparison between ResNets trained using a warm start and a random initialization on CIFAR-10. Blue lines are models trained on 50% of CIFAR-10 for 350 epochs then trained on 100% of the data for a further 350 epochs. Orange lines are models trained on 100% of the data from the start. The two procedures produce similar training performance but differing test performance.

the network trained on the same data but with a new random initialization. Problematically, this phenomenon incentivizes performance-focused researchers and engineers to constantly retrain models from scratch, at potentially enormous financial and environmental cost [4]. This is an example of "Red AI" [5], disregarding resource consumption in pursuit of raw predictive performance.

The warm-start phenomenon has implications for other situations as well. In active learning, for example, unlabeled samples are abundant but labels are expensive: the goal is to identify maximally-informative data to have labeled by an oracle and integrated into the training set. It would be time efficient to simply warm-start optimization each time new samples are appended to the training set, but such an approach seems to damage generalization in deep neural networks. Although this phenomenon has not received much direct attention from the research community, it seems to be common practice in deep active learning to retrain from scratch after every query step [6, 7]; popular deep active learning repositories on Github randomly reinitialize models after every selection. [8, 9].

The ineffectiveness of warm-starting has been observed anecdotally in the community, but this paper seeks to examine its properties closely in controlled settings. Note that the findings in this paper are not inconsistent with extensive work on unsupervised pre-training [10, 11] and transfer learning in the small-data and "few shot" regimes [12–15]. Rather here we are examining how to accelerate training in the large-data supervised setting in a way consistent with expectations from convex problems.

This article is structured as follows. Section 2 examines the generalization gap induced by warm-starting neural networks. Section 3 surveys approaches for improving generalization in deep learning, and shows that these techniques do not resolve the problem. In Section 4, we describe a simple trick that overcomes this pathology, and report on experiments that give insights into its behavior in batch online learning and pre-training scenarios. We defer our discussion of related work to Section 5, and include a statement on broad impacts in Section 6.

## 2   Warm Starting Damages Generalization

In this section we provide empirical evidence that warm starting consistently damages generalization performance in neural networks. We conduct a series of experiments across several different architectures, optimizers, and image datasets. Our goal is to create simple, reproducible settings in which the warm-starting phenomenon is observed.

### 2.1   Basic Batch Updating

Here we consider the simplest case of warm-starting, in which a single training dataset is partitioned into two subsets that are presented sequentially. In each series of experiments, we randomly segment the training data into two equally-sized portions. The model is trained to convergence on the first half, then is trained on the union of the two batches, i.e., 100% of the data. This is repeated for three classifiers: ResNet-18 [16], a multilayer perceptron (MLP) with three layers and tanh activations, and logistic regression. Models are optimized using either stochastic gradient descent (SGD) or the Adam variant of SGD [17], and are fitted to the CIFAR-10, CIFAR-100, and SVHN image data. All models are trained using a mini-batch size of 128 and a learning rate of 0.001, the smallest learning rate used in the learning schedule for fitting state-of-the-art ResNet models [16]. The effect of these parameters is investigated in Section 3. Presented results are on a held-out, randomly-chosen third of available data.

| CIFAR-10 | RESNET SGD | RESNET ADAM | MLP SGD | MLP ADAM | LR SGD | LR ADAM |
|---|---|---|---|---|---|---|
| RANDOM INIT | 56.2 (1.0) | 78.0 (0.6) | 39.0 (0.2) | 39.4 (0.1) | 40.5 (0.6) | 33.8 (0.6) |
| WARM START | 51.7 (0.9) | 74.4 (0.9) | 37.4 (0.2) | 36.1 (0.3) | 39.6 (0.2) | 33.3 (0.2) |
| **SVHN** | | | | | | |
| RANDOM INIT | 89.4 (0.1) | 93.6 (0.2) | 76.5 (0.3) | 76.7 (0.4) | 28.0 (0.2) | 22.4 (1.3) |
| WARM START | 87.5 (0.7) | 93.5 (0.4) | 75.4 (0.1) | 69.4 (0.6) | 28.0 (0.3) | 22.2 (0.9) |
| **CIFAR-100** | | | | | | |
| RANDOM INIT | 18.2 (0.3) | 41.4 (0.2) | 10.3 (0.2) | 11.6 (0.2) | 16.9 (0.18) | 10.2 (0.4) |
| WARM START | 15.5 (0.3) | 35.0 (1.2) | 9.4 (0.0) | 9.9 (0.1) | 16.3 (0.28) | 9.9 (0.3) |

Table 1: Validation percent accuracies for various optimizers and models for warm-started and randomly initialized models on indicated datasets. We consider an 18-layer ResNet, three-layer multilayer perceptron (MLP), and logistic regression (LR).

Our results (Table 1) indicate that generalization performance is damaged consistently and significantly for both ResNets and MLPs. This effect is more dramatic for CIFAR-10, which is considered relatively challenging to model (requiring, e.g., data augmentation), than for SVHN, which is considered easier. Logistic regression, which enjoys a convex loss surface, is not significantly damaged by warm starting for any datasets. Figure 10 in the Appendix extends these results and shows that the gap is inversely proportional to the fraction of data available in the first round of training.

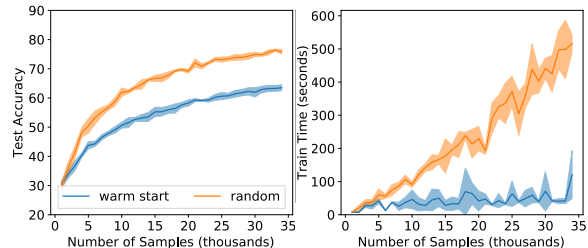

Figure 2: An online learning experiment for CIFAR-10 data using a ResNet. The horizontal axis shows the total number of samples in the training set available to the learner. The generalization gap between warm-started and randomly-initialized models is significant.

This result is surprising. Even though MLP and ResNet optimization is non-convex, conventional intuition suggests that the warm-started solution should be close to the full-data solution and therefore a good initialization. One view on pre-training is that the initialization is a "prior" on weights; we often view prior distributions as arising from inference on old (or hypothetical) data and so this sort of pre-training should always be helpful. The generalization gap shown here creates a computational burden for real-life machine learning systems that must be retrained from scratch to perform well, rather than initialized from previous models. First-round results for Table 1 are in Appendix Table 2.

## 2.2 Online Learning

A common real-world setting involves data that are being provided to the machine learning system in a stream. At every step, the learner is given $k$ new samples to append to its training data, and it updates its hypothesis to reflect the larger dataset. Financial data, social media data, and recommendation systems are common examples of scenarios where new samples are constantly arriving. This paradigm is simulated in Figure 2, where we supply CIFAR-10 data, selected randomly without replacement, in batches of 1,000 to an 18-layer ResNet. We examine two cases: 1) where the model is retrained from scratch after each batch, starting from a random initialization, and 2) where the model is trained to convergence starting from the parameters learned in the previous iteration. In both cases, the models are optimized with Adam, using an initial learning rate of 0.001. Each was run five times with different random seeds and validation sets composed of a random third of available data, reinitializing Adam's parameters at each step of learning.

Figure 2 shows the trade-off between these two approaches. On the right are the training times: clearly, starting from the previous model is preferable and has the potential to vastly reduce computational costs and wall-clock time. However, as can be seen on the left, generalization performance is worse in the warm-started situation. As more data arrive, the gap in validation accuracy increases substantially. Means and standard deviations across five runs are shown. Although this work focuses on image data, we find consistent results with other dataset and architecture choices (Appendix Figure 13).

## 3 Conventional Approaches

The design space for initializing and training deep neural network models is very large, and so it is important to evaluate whether there is some known method that could be used to help warm-started training find good solutions. Put another way, a reasonable response to this problem is "Did you see whether $X$ helped?" where $X$ might be anything from batch normalization [18] to increasing mini-batch size [19]. This section tries to answer some of these questions and further empirically probe the warm-start phenomenon. Unless otherwise stated, experiments in this section use a ResNet-18 model trained using SGD with a learning rate of 0.001 on CIFAR-10 data. All experiments were run five times to report means and standard deviations. No experiments in this paper use data augmentation or learning rate schedules, and all validation sets are a randomly-chosen third of the training data.

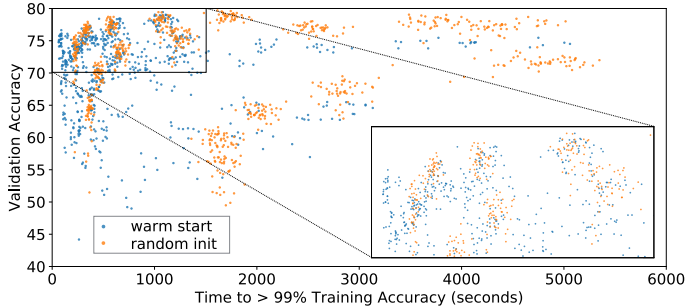

Figure 3: A comparison between ResNets trained from both a warm start and a random initialization on CIFAR-10 for various hyperparameters. Orange dots are randomly-initialized models and blue dots are warm-started models. Warm-started models that perform roughly as well as randomly-initialized models offer no benefit in terms of training time.

## 3.1 Is this an effect of batch size or learning rate?

One might reasonably ask whether or not there exist *any* hyperparameters that close the generalization gap between warm-started and randomly-initialized models. In particular, can setting a larger learning rate at either the first or second round of learning help the model escape to regions that generalize better? Can shrinking the batch size inject stochasticity that might improve generalization [20, 21]?

Here we again consider a warm-started experiment of training on 50% of CIFAR-10 until convergence, then training on 100% of CIFAR-10 using the initial round of training as an initialization. We explore all combinations of batch sizes $\{16, 32, 64, 128\}$, and learning rates $\{0.001, 0.01, 0.1\}$, varying them across the three rounds of training. This allows for the possibility that there exist different hyperparameters for the first stage of training that are better when used with a different set after warm-starting. Each combination is run with three random initializations.

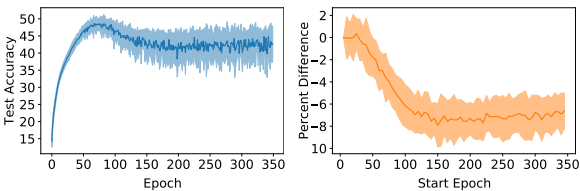

Figure 4: **Left:** Validation accuracy as training progresses on 50% of CIFAR-10. **Right:** Validation accuracy damage, as percentage difference from random initialization, after training on 100% of the data. Each warm-started model was initialized by training on 50% of CIFAR data for the indicated number of epochs.

Figure 3 visualizes these results. Every resulting 100% model is shown from all three initializations and all combinations, with color indicating whether it was a random initialization or a warm-start. The horizontal axis shows the time to completion, excluding the pre-training time, and the vertical axis shows the resulting validation performance.

Interestingly, we do find warm-started models that perform as well as randomly-initialized models, but they are unable to do so while benefiting from their warm-started initialization. The training time for warm-started ResNet models that generalize as well as randomly-initialized models is roughly the same as those randomly-initialized models. That is, there is no computational benefit to using these warm-started initializations. It is worth noting that this plot does not capture the time or energy required to identify hyperparameters that close the generalization gap; such hyperparameter searches are often the culprit in the resource footprint of deep learning [5]. Wall-clock time is measured by assigning every model identical resources, consisting of 50GB of RAM and an NVIDIA Tesla P100 GPU.

This increased fitting time occurs because warm-started models, when using hyperparameters that generalize relatively well, seem to "forget" what was learned in the first round of training. Appendix Figure 11 provides evidence this phenomenon by computing the Pearson correlation between the weights of converged warm-started models and their initialization weights, again across various choices for learning rate and batch size, and comparing it to validation accuracy. Models that generalize well have little correlation with their initialization—there is a trend downward in accuracy with increasing correlation—suggesting that they have forgotten what was learned in the first round of training. Conversely, a similar plot for logistic regression shows no such relationship.

## 3.2 How quickly is generalization damaged?

One surprising result in our investigation is that only a small amount of training is necessary to damage the validation performance of the warm-started model. Our hope was that warm-starting success might be achieved by switching from the 50% to 100% phase before the first phase of training was completed. We fit a ResNet-18 model on 50% of the training data, as before, and checkpointed its parameters every five epochs. We then took each of these checkpointed models and used them as an initialization for training on 100% of those data. As shown in Figure 4, generalization is damaged even when initializing from parameters obtained by training on incomplete data for only a few epochs.

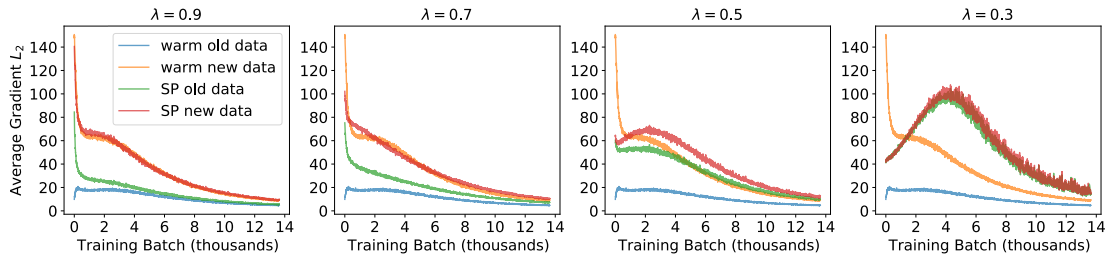

Figure 5: A two-phase experiment like those in Sections 2 and 3, where a ResNet is trained on 50% of CIFAR-10 and is then given the remainder in the second round of training. Here we examine the average gradient norms separately corresponding to the initial 50% of data and the second 50% for models that are either warm-started or initialized with the shrink and perturb (SP) trick. Notice that in warm-started models, there is a drastic gap between these gradient norms. Our proposed trick balances these respective magnitudes while still allowing models to benefit from their first round of training; i.e they fit training data much quicker than random initializations.

### 3.3 Is regularization helpful?

A common approach for improving generalization is to include a regularization penalty. Here we investigate three different approaches to regularization: 1) basic $L_2$ weight penalties [22], 2) confidence-penalized training [23], and 3) adversarial training [24]. We again take a ResNet fitted to 50% of available training data and use its parameters to warm-start learning on 100% of the data. We apply regularization in both rounds of training, and while it is helpful, regularization does not resolve the generalization gap induced by warm starting. Appendix Table 3 shows the result of these experiments for indicated regularization penalty sizes. Our experiments show that applying the same amount of regularization to randomly-initialized models still produces a better-generalizing classifier.

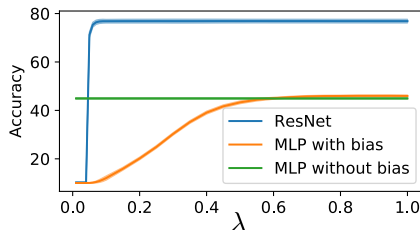

Figure 6: We fit a ResNet and MLP (with and without bias nodes) to CIFAR-10 and measure performance as a function of the shrinkage parameter $\lambda$.

## 4 Shrink, Perturb, Repeat

While the presented conventional approaches do not remedy the warm-start problem, we have identified a remarkably simple trick that efficiently closes the generalization gap. At each round of training $t$, when new samples are appended to the training set, we propose initializing the network's parameters by shrinking the weights found in the previous round of optimization towards zero, then adding a small amount of parameter noise. Specifically, we initialize each learnable parameter $\theta_i^t$ at training round $t$ as $\theta_i^t \leftarrow \lambda\theta_i^{t-1} + p^t$, where $p^t \sim \mathcal{N}(0, \sigma^2)$ and $0 < \lambda < 1$.

**Shrinking weights preserves hypotheses.** For network layers that use ReLU nonlinearities, shrinking parameters preserves the relative activation at each layer. If bias terms and batch normalization are not used, the output of every layer is a scaled version of its non-shrunken counterpart. In the last layer, which usually consists of a linear transformation followed by a softmax nonlinearity, shrinking parameters can be interpreted as increasing the entropy of the output distribution, effectively diminishing the model's confidence. For no-bias, no-batchnorm ReLU models, while shrinking weights does not necessarily preserve the output $f_\theta(x)$ they parametrize, it does preserve the learned hypothesis, i.e. $\arg\max f_\theta(x)$; a simple proof is provided for completeness as Proposition 1 in the Appendix.

For more sophisticated architectures, this property largely still holds: Figure 6 shows that for a ResNet, which includes batch normalization, only extreme amounts of shrinking are able to damage classifier performance. This is because batch normalization's internal estimates of mean and variance can compensate for the rescaling caused by weight shrinking. Even for a ReLU MLP that includes bias nodes, performance is surprisingly resilient to shrinking; classifier damage is done only for $\lambda < 0.6$ in Figure 6. Separately, note that when internal network layers instead use sigmoidal activations, shrinking parameters moves them further from saturating regions, allowing the model to more easily learn from new data.

**Shrink-perturb balances gradients.** Figure 5 shows a visualization of average gradients during the second of a two-phase training procedure for a ResNet on CIFAR-10, like those discussed in Sections 2 and 3. We plot the second phase of training, where gradient magnitudes are shown separately for the two halves of the dataset. For this experiment models are optimized with SGD, using a small learning rate to zoom in on this effect. Outside of this plot, experiments in this section use the Adam optimizer.

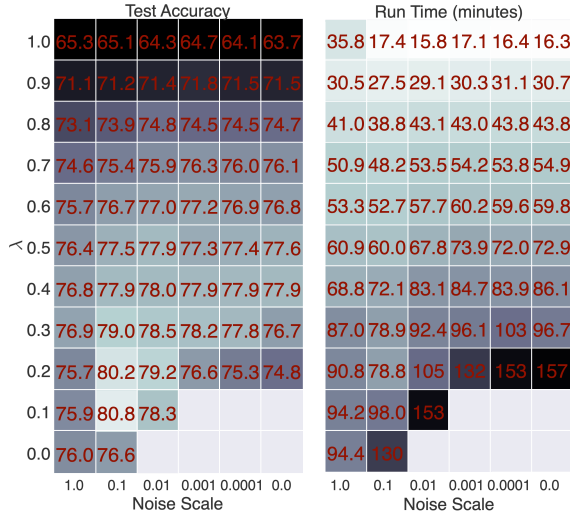

| $\lambda$ | \ Test Accuracy | | | | | |
|---|---|---|---|---|---|---|
| 1.0 | 65.3 | 65.1 | 64.3 | 64.7 | 64.1 | 63.7 |
| 0.9 | 71.1 | 71.2 | 71.4 | 71.8 | 71.5 | 71.5 |
| 0.8 | 73.1 | 73.9 | 74.8 | 74.5 | 74.5 | 74.7 |
| 0.7 | 74.6 | 75.4 | 75.9 | 76.3 | 76.0 | 76.1 |
| 0.6 | 75.7 | 76.7 | 77.0 | 77.2 | 76.9 | 76.8 |
| 0.5 | 76.4 | 77.5 | 77.9 | 77.3 | 77.4 | 77.6 |
| 0.4 | 76.8 | 77.9 | 78.0 | 77.9 | 77.9 | 77.9 |
| 0.3 | 76.9 | 79.0 | 78.5 | 78.2 | 77.8 | 76.7 |
| 0.2 | 75.7 | 80.2 | 79.2 | 76.6 | 75.3 | 74.8 |
| 0.1 | 75.9 | 80.8 | 78.3 | | | |
| 0.0 | 76.0 | 76.6 | | | | |
| Noise Scale | 1.0 | 0.1 | 0.01 | 0.001 | 0.0001 | 0.0 |

| Run Time (minutes) | | | | | | |
|---|---|---|---|---|---|---|
| 35.8 | 17.4 | 15.8 | 17.1 | 16.4 | 16.3 |
| 30.5 | 27.5 | 29.1 | 30.3 | 31.1 | 30.7 |
| 41.0 | 38.8 | 43.1 | 43.0 | 43.8 | 43.8 |
| 50.9 | 48.2 | 53.5 | 54.2 | 53.8 | 54.9 |
| 53.3 | 52.7 | 57.7 | 60.2 | 59.6 | 59.8 |
| 60.9 | 60.0 | 67.8 | 73.9 | 72.0 | 72.9 |
| 68.8 | 72.1 | 83.1 | 84.7 | 83.9 | 86.1 |
| 87.0 | 78.9 | 92.4 | 96.1 | 103 | 96.7 |
| 90.8 | 78.8 | 105 | 122 | 153 | 157 |
| 94.2 | 98.0 | 153 | | | |
| 94.4 | 130 | | | | |

Noise Scale: 1.0  0.1  0.01  0.001  0.0001  0.0

Figure 8: Model performance as a function of $\lambda$ and $\sigma$. Numbers indicate the average final performance and total train time for online learning experiments where ResNets are provided CIFAR-10 samples in sequence, 1,000 per round, and trained to convergence at each round. Note that the bottom left of this plot corresponds to pure random initializing while the top right corresponds to pure warm starting. **Left**: Validation accuracy tends to improve with more aggressive shrinking. Adding noise often improves generalization. **Right**: Model train times increase with decreasing values of $\lambda$. This is expected, as decreasing $\lambda$ widens the gap between shrink-perturb parameters and warm-started parameters. Noise helps models train more quickly. Unlabeled boxes correspond to initializations too small for the model to reliably learn.

For warm-started models, gradients from new, unseen data tend to be much larger magnitude than those from data the model has seen before. These imbalanced gradient contributions are known to be problematic for optimization in mutli-task learning scenarios [25], and suggest that under warm-started initializations the model does not learn in the same way as it would with randomly-initializied training [26]. We find that remedying this imbalance without damaging what the model has already learned is key to efficiently resolving the generalization gap studied in this article.

Shrinking the model's weights increases its loss, and correspondingly increases the magnitude of the gradient induced even by samples that have already been seen. Preposition 1 shows that in an $L$-layer ReLU network without bias nodes or batch normalization, shrinking weights by $\lambda$ shrinks softmax inputs by $\lambda^L$, rapidly increasing the entropy of the softmax distribution and the cross-entropy loss. As shown in Figure 5, the loss increase caused by shrink perturb trick is able to balance gradient contributions between previously unseen samples and data on which the model has already been trained.

The success of the shrink and perturb trick lies in its ability to standardize gradients while preserving learned hypotheses. We could instead normalize gradient contributions by, for example, adding a significant amount of parameter noise, but this also damages the learned function. Consequently, this strategy drastically increases training time without fully closing the warm-start generalization gap (Appendix Table 4). As an alternative to shrinking all weights, we could try to increase the entropy of the output distribution by shrinking only parameters in the last layer (Appendix Figure 14), or by regularizing the model's confidence while training (Appendix Table 3), but these are unable to resolve the warm-start problem. For sophisticated architectures especially, we find it is important to holistically modify parameters before training on new data.

The perturbation step, adding noise after shrinking, improves both training time and generalization performance. The trade-off between relative values of $\lambda$ and $\sigma$ is studied in Figure 8. Note that in this figure, and in this section generally, we refer to the "noise scale" rather than to $\sigma$. In practice, we add noise by adding parameters from a scaled, randomly-initialized network, to compensate for the fact that many random initialization schemes use different variances for different kinds of parameters.

Figure 7 demonstrates the effectiveness of this trick. Like before, we present a passive online learning experiment where 1,000 CIFAR-10 samples are supplied to a ResNet in sequence. At each round we can either reinitialize network parameters from scratch or warm start, initializing them to

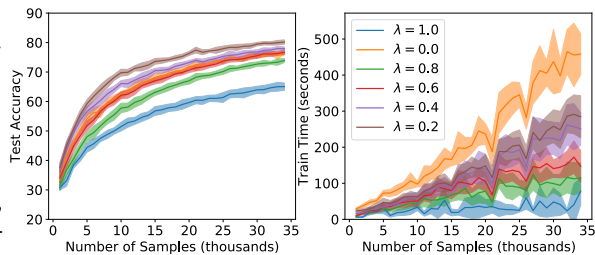

Figure 7: An online learning experiment varying $\lambda$ and keeping the noise scale fixed at $0.01$. Note that $\lambda = 1$ corresponds to fully-warm-started initializations and $\lambda = 0$ corresponds to fully-random initializations. The proposed trick with $\lambda = 0.6$ performs identically to randomly initializing in terms of validation accuracy, but trains much more quickly. Interestingly, smaller values of $\lambda$ are even able to outperform random initialization while still training faster.

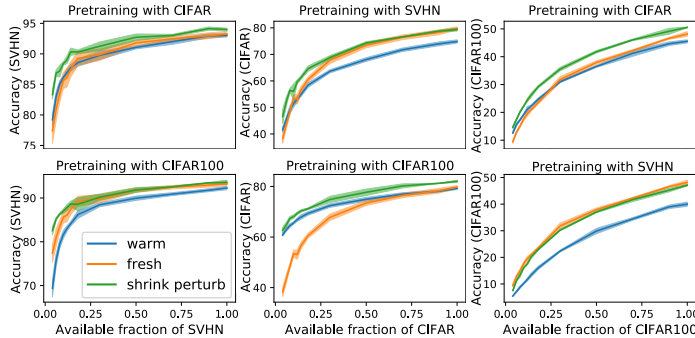

Figure 9: Pre-trained models fitted to a varying fraction of the indicated dataset. We compare these warm-started, pre-trained models to randomly initialized and shrink-perturb initialized counterparts, trained on the same fraction of target data. The relative performance of warm-starting and randomly initializing varies, but shrink-perturb performs at least as well as the best strategy.

those found in the previous round of optimization. As expected, we see that warm-started models train faster but generalize worse. However, if we instead initialize parameters using the shrink and perturb trick, we are able to both close this generalization gap and significantly speed up training. Appendix Sections 8.2.1–8.2.6 present extensive results varying $\lambda$ and noise scale, experimenting with dataset type, model architecture, and $L_2$ regularization, all showing the same overall trend. Indeed, we notice that shrink-perturb parameters that better balance gradient contributions better remedy the warm-start problem. That said, we find that one does not need to shrink very aggressively to adequately enough correct gradients and efficiently close the warm-start generalization gap.

## 4.1 The shrink and perturb trick and regularization

Exercising the shrink and perturb trick at every step of SGD would be very similar to applying an aggressive, noisy $L_2$ regularization. That is, shrink-perturbing every step of optimization yields the SGD update $\theta_i \leftarrow \lambda(\theta_i + \eta \frac{\partial L}{\partial \theta_i}) + p$ for loss $L$, weight $\theta_i$, and learning rate $\eta$, making the shrinkage term $\lambda$ behave like a weight decay parameter. It is natural to ask, then, how does this trick compare with weight decay? Appendix Figure 12 shows that in non-warm-started environments, where we just have a static dataset, the iterative application of the shrink-perturb trick results in marginally improved performance. These experiments fit a ResNet to convergence on 100% of CIFAR-10 data, then shrink and perturb weights before repeating the process, resulting in a modest performance improvement. We can conclude that the shrink-perturb trick has two benefits. Most significantly, it allows us to quickly fit high-performing models in sequential environments without having to retrain from scratch. Separately, it offers a slight regularization benefit, which in tandem with the first property sometimes allows shrink-perturb models to generalize even better than randomly-initialized models.

This $L_2$ regularization benefit is not enough to explain the success of the shrink-perturb trick. As Appendix Table 3 demonstrates, $L_2$-regularized models are still vulnerable to the warm-start generalization gap. Appendix Sections 8.2.5 and 8.2.6 show that we are able to mitigate this performance gap with the shrink and perturb trick even when models are being aggressively regularized (regularization penalties any larger prevent networks from being able to fit the training data) with weight decay.

## 4.2 The shrink and perturb trick and pre-training

Despite successes on a variety of tasks, deep neural networks still generally require large training sets to perform well. For problems where only limited data are available, it has become popular to warm-start learning using parameters from training on a different but related problem [14, 27]. Transfer and "few-shot" learning in this form has seen success in computer vision and NLP [28].

The experiments we perform here, however, imply that when the second problem is not data-limited, this transfer learning approach deteriorates model quality. That is, at some point, the pre-training transfer learning approach is similar to warm-starting under domain shift, and generalization should suffer.

We demonstrate this phenomenon by first training a ResNet-18 to convergence on one dataset, then using that solution to warm-start a model trained on a varying fraction of another dataset. When only a small portion of target data are used, this is essentially the same as the pre-training transfer learning approach. As the proportion increases, the problem turns into what we have described here as warm starting. Figure 9 shows the result of this experiment, and it appears to support our intuition. Often, when the second dataset is small, warm starting is helpful, but there is frequently a crossover point where better generalization would be achieved by training from scratch on that fraction of the target data. Sometimes, when source and target datasets are dissimilar, it would better to randomly initialize regardless of the amount of target data available.

The exact point at which this crossover occurs (and whether it happens at all) depends not just on model type but also on the statistical properties of the data in question; it cannot be easily predicted. We find that shrink-perturb initialization, however, allows us to avoid having to make such a prediction: shrink-perturbed models perform at least as well as warm-started models when pre-training is the most performant strategy and as well as randomly-initialized models when it is better to learn from scratch. Figure 9 displays this effect for $\lambda = 0.3$ and noise scale 0.0001. Comprehensive shrink-perturb settings are presented for this scenario in Appendix Section 8.2.7, all showing similar results.

## 5  Discussion and Research Surrounding the Warm Start Problem

Warm-starting and online learning are well understood for convex models like linear classifiers [29] and SVMs [30, 31]. Excluding the shrink-perturb trick, it does not appear that generally applicable techniques exist for deep neural networks that do not damage generalization, so models are typically retrained from scratch [6, 32].

There has been a variety of work in closely related areas, however. For example, in analyzing "critical learning periods," researchers show that a network initially trained on blurry images then on sharp images is unable to perform as well as one trained from scratch on sharp images, drawing a parallel between human vision and computer vision [26]. We show that this phenomenon is more general, with test performance damaged even when first and second datasets are drawn from identical distributions.

**Initialization.**  The problem of warm starting is closely related to the rich literature on initialization of neural network training "from scratch". Indeed, new insights into what makes an effective initialization have been critical to the revival of neural networks as machine learning models. While there have been several proposed methods for initialization [33, 34, 10, 35, 36], this body of literature primarily concerns itself with initializations that are high-quality in the sense that they allow for quick and reliable model training. That is, these methods are typically built with training performance in mind rather than generalization performance.

Work relating initialization to generalization suggests that networks whose weights have moved far from their initialization are less likely to generalize well compared with ones that have remained relatively nearby [37]. Here we have shown with experimental results that warm-started networks that have *less* in common with their initializations seem to generalize better than those that have more (Appendix Figure 11). So while it is not surprising that there exist initializations that generalize poorly, it is surprising that warm starts are in that class. Still, before retraining, our proposed solution brings parameters closer their initial values than they would be if just warm starting, suggesting some relationship between generalization and distance from initialization.

**Generalization.**  The warm-start problem is fundamentally about generalization performance, which has been extensively studied both theoretically and empirically within the context of deep learning. These articles have investigated generalization by studying classifier margin [38, 39], loss geometry [40, 19, 41], and measurements of complexity [42, 43], sensitivity [44], or compressiblity [45].

These approaches can be seen as attempting to measure the intricacy of the hypothesis learned by the network. If two models are both consistent for the same training data, the one with the less complicated concept is more likely to generalize well. We know that networks trained with SGD are implicitly regularized [20, 21], suggesting that standard training of neural networks incidentally finds low-complexity solutions. It's possible, then, that the initial round of training disqualifies solutions that would most naturally explain the general problem of interest. If so, by balancing gradient contributions, the shrink and perturb trick seems to make these solutions accessible again.

**Pre-training.**  As previously discussed, the warm-start problem is very similar to the idea of unsupervised and supervised pre-training [46, 11, 10, 47]. Under that paradigm, learning where limited labeled data are available is aided by first training on related data. The warm start problem, however, is not about limited labeled data in the second round of training. Instead, the goal of warm starting is to hasten the time required to fit a neural network by initializing using a similar supervised problem without damaging generalization. Our results suggest that while warm-starting is beneficial when labeled data are limited, it actually damages generalization to warm-start in data-rich situations.

**Concluding thoughts.**  This article presented the challenges of warm-starting neural network training and proposed a simple and powerful solution. While warm-starting is a problem that the community seems somewhat aware of anecdotally, it does not seem to have been directly studied. We believe that this is a major problem in important real-life tasks for which neural networks are used, and it speaks directly to the resources consumed by training such models.

# 6   Broader Impact

The shrink and perturb trick allows models to be efficiently updated without sacrificing generalization performance. In the absence of this method, achieving best-possible performance requires neural networks to be randomly-initialized each time new data are appended to the training set. As mentioned earlier, this requirement can cost significant computational resources, and as a result, is partially responsible for the deleterious environmental ramifications studied in recent years [4, 5].

Additionally, the enormous computational expense of retraining models from scratch disproportionately burdens research groups without access to abundant computational resources. The shrink and perturb trick lowers this barrier, democratizing participation in online learning, active learning, and pre-training research with neural networks.

# 7   Funding Disclosure and Competing Interests

This work was partially funded by NSF IIS-2007278 and by a Siemens FutureMakers graduate student fellowship. RPA is on the board of directors at Cambridge Machines Ltd. and is a scientific advisor to Manifold Bio.

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
