[Supplementary Material]

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

# 8 Appendix

**Proposition 1.** *Consider a neural network $f_\theta$ trained to predict one of $k$ classes and paramatrized by weight matrices $\theta = (W_1, W_2, .., W_L)$. Using the ReLU nonlinearity $\sigma(z) = \max(0, z)$ and $\mathrm{softmax}(z)_i = e^{z_i}/\sum_{j=1}^{k} e^{z_j}$, let $f_\theta(x) = \mathrm{softmax}(W_L \cdot \sigma(W_{L-1} \cdot .. \cdot \sigma(W_2 \cdot \sigma(W_1 \cdot x))))$. Then, for $\lambda > 0$ and input $x$, $\mathrm{argmax}\, f_\theta(x) = \mathrm{argmax}\, f_{\lambda\theta}(x)$.*

*Proof.* Observe that $\sigma(\lambda z) = \lambda\sigma(z) \;\; \forall \lambda > 0$. Then,

$$
\begin{aligned}
\mathrm{argmax}\, f_{\lambda\theta}(x) &= \mathrm{argmax}\, \mathrm{softmax}(\lambda W_L \cdot \sigma(\lambda W_{L-1} \cdot .. \cdot \sigma(\lambda W_2 \cdot \sigma(\lambda W_1 \cdot x)))) \\
&= \mathrm{argmax}\, \mathrm{softmax}(\lambda^L W_L \cdot \sigma(W_{L-1} \cdot .. \cdot \sigma(W_2 \cdot \sigma(W_1 \cdot x)))) \\
&= \mathrm{argmax}\, \mathrm{softmax}(W_L \cdot \sigma(W_{L-1} \cdot .. \cdot \sigma(W_2 \cdot \sigma(W_1 \cdot x)))) \\
&= \mathrm{argmax}\, f_\theta(x)
\end{aligned}
$$

$\square$

## 8.1 Appendix Tables

Table 2: Validation percent accuracies for various optimizers and models for the first round of warm-started training, i.e. training on half of the training data available in Table 1. We consider an 18-layer ResNet, three-layer multilayer perceptron (MLP), and logistic regression (LR) as our classifiers. Validation sets are a randomly-chosen third of the training data. Standard deviations are indicated parenthetically.

|  | RESNET SGD | RESNET ADAM | MLP SGD | MLP ADAM | LR SGD | LR ADAM |
|---|---|---|---|---|---|---|
| CIFAR-10 | 41.7 (7.9) | 70.5 (1.6) | 37.2 (0.2) | 36.0 (0.2) | 37.9 (0.2) | 31.8 (0.7) |
| SVHN | 85.9 (0.3) | 92.3 (0.2) | 72.5 (0.4) | 67.5 (0.3) | 27.1 (0.3) | 22.2 (0.7) |
| CIFAR-100 | 10.6 (1.6) | 31.5 (0.7) | 10.3 (0.2) | 10.5 (0.3) | 15.4 (0.21) | 9.3 (0.3) |

Table 3: Validation percent accuracies for various optimizers and models for the first round of warm-started training, i.e. training on half of the training data available in Table 1. We consider an 18-layer ResNet, three-layer multilayer perceptron (MLP), and logistic regression (LR) as our classifiers. Validation sets are a randomly-chosen third of the training data. Standard deviations are indicated parenthetically.

| **L2** | $1 \times 10^{-1}$ | $1 \times 10^{-2}$ | $1 \times 10^{-3}$ | $1 \times 10^{-4}$ |
|---|---|---|---|---|
| RI | 72.7 (4.2) | 55.4 (2.7) | 54.6 (2.4) | 55.1 (3.4) |
| WS | 63.9 (6.4) | 51.2 (2.7) | 50.5 (1.8) | 50.4 (1.3) |
| **ADVERSARIAL** | | | | |
| RI | 54.8 (1.3) | 55.1 (1.5) | 55.3 (1.4) | 55.6 (0.9) |
| WS | 52.4 (1.0) | 52.6 (1.5) | 52.7 (1.2) | 50.4 (1.4) |
| **CONFIDENCE** | | | | |
| RI | 53.1 (1.9) | 55.8 (1.3) | 55.4 (1.2) | 55.9 (1.4) |
| WS | 50.3 (0.7) | 50.0 (3.8) | 51.2 (1.2) | 49.3 (1.2) |

Table 4: Validation accuracies and warm-started model train times (minutes). Adding noise at the indicated standard deviations improves generalization, but not to the point of performing as well as randomly-initialized models. Better-generalizing warm-started models take even more time to train than their randomly-initialized peers, which on average achieve 55.2% accuracy in 34.0 minutes.

|  | $1 \times 10^{-2}$ | $1 \times 10^{-3}$ | $1 \times 10^{-4}$ | $1 \times 10^{-5}$ | 0 |
|---|---|---|---|---|---|
| Accuracy | 54.4 (0.9) | 53.5 (1.0) | 52.9 (1.0) | 49.9 (1.6) | 50.8 (1.8) |
| Train Time | 165.3 (3.9) | 38.0 (1.33) | 16.5 (1.3) | 14.6 (91.0) | 13.6 (0.4) |

Table 5: Validation percent accuracies for various datasets for last layer only warm-starting (LL), last layer warm starting followed by full network training (LL+WS), warm started (WS) and randomly initialized (RI) models on various indicated datasets.

|  | LL | LL+WS | WS | RI |
|---|---|---|---|---|
| CIFAR-10 | 48.8 (1.8) | 50.9 (1.5) | 52.5 (0.3) | 56.0 (1.2) |
| SVHN | 86.0 (0.6) | 88.2 (0.2) | 87.5 (0.7) | 89.4 (0.1) |
| CIFAR-100 | 16.4 (0.5) | 16.5 (0.6) | 15.5 (0.3) | 18.2 (0.3) |

## 8.2 Appendix Figures

Figure 10: Warm-started ResNet generalization as a function of the fraction of total data available in the first round of training. Models are trained on the indicated fraction of CIFAR-10 training data until convergence, then trained again on 100% of CIFAR-10 data to produce this figure. When the initial data used to warm-start training more overlaps with the second round of training data, the generalization gap is less severe.

Figure 11: Validation accuracy as a function of the correlation between the warm-start initialization and the solution found after training for a large number of hyperparameter settings. **Left**: Warm-started logistic regressors often remember their initialization. **Right**: Warm-started ResNets that perform well do not retain much information from the initial round of training.

Figure 12: The result of fitting a ResNet on 100% of CIFAR-10 to convergence for twenty rounds and applying the shrink-perturb trick after each. Here we show four versions of that experiment for the indicated $\lambda$ and a noise scaling of 0.01. Iteratively application has a slight regularization effect.

Figure 13: An online learning experiment using a two-layer bidirectional RNN trained on the IMDB movie review sentiment classification dataset. Samples are supplied iid in batches of 1,000. Like with other experiments, warm starting ($\lambda = 1$) performs significantly worse than randomly initializing ($\lambda = 0$). Shrink-perturb initialization closes this generalization gap.

Figure 14: An online learning experiment using a ResNet on CIFAR-10 data. Data are supplied iid in batches of 1,000. Here, instead of shrinking and perturbing every weight in the model, we modify only those in the last layer. Models modified this way, unlike the shrink-perturb trick we present, which modifies every parameter in the network, these retrained models are unable to outperform even purely warm-started models.

### 8.2.1 Batch Online Learning Results for a ResNet-18 on CIFAR-10

This section shows results of a shrink and perturb online learning experiment with a ResNet-18 on CIFAR-10 data, iteratively supplying batches of 1,000 to the model and training it to convergence.

|       | Test Accuracy |      |      |      |      |      | Run Time (minutes) |      |      |      |      |      |
|-------|------|------|------|------|------|------|------|------|------|------|------|------|
| **1.0** | 65.3 | 65.1 | 64.3 | 64.7 | 64.1 | 63.7 | 35.8 | 17.4 | 15.8 | 17.1 | 16.4 | 16.3 |
| **0.8** | 73.1 | 73.9 | 74.8 | 74.5 | 74.5 | 74.7 | 41.0 | 38.8 | 43.1 | 43.0 | 43.8 | 43.8 |
| **0.6** | 75.7 | 76.7 | 77.0 | 77.2 | 76.9 | 76.8 | 53.3 | 52.7 | 57.7 | 60.2 | 59.6 | 59.8 |
| **0.4** | 76.8 | 77.9 | 78.0 | 77.9 | 77.9 | 77.9 | 68.8 | 72.1 | 83.1 | 84.7 | 83.9 | 86.1 |
| **0.2** | 75.7 | 80.2 |      |      |      |      | 90.8 | 78.8 |      |      |      |      |
| **0.0** | 76.0 | 76.6 |      |      |      |      | 94.4 | 130  |      |      |      |      |

λ (rows), Noise Scale (columns: 1.0, 0.1, 0.01, 0.001, 0.0001, 0.0)

Figure 15: Average performance resulting from using the shrink and perturb trick with varying choices for λ and noise scale. Final accuracies and train times. Missing numbers correspond to initializations that were too small to be trained. The bottom left entry is a pure random initialization while the top right is a pure warm start.

Figure 16: Complete learning curves corresponding to each entry of Figure 15, where λ = 0 is warm starting and λ = 1 is randomly initializing (plus the indicated noise amount).

### 8.2.2 Batch Online Learning Results for a ResNet-18 on SVHN

Here we show an online learning experiment with a ResNet-18 on SVHN data, iteratively supplying batches of 1,000 to the model and training it to convergence.

Figure 17: Average fianl accuracies and train times when using the shrink and perturb trick with varying choices for $\lambda$ and noise scale. Missing numbers correspond to initializations that were too small to be trained. The bottom left entry is a pure random initialization while the top right is a pure warm start.

Figure 18: Complete learning curves corresponding to each entry of Figure 17, where $\lambda = 0$ is warm starting and $\lambda = 1$ is randomly initializing (plus the indicated noise amount).

### 8.2.3 Batch Online Learning Results for an MLP on CIFAR-10 (no batch normalization)

Here we show an online learning experiment, training an MLP consisting of three layers, ReLU activations, and 100-dimensional hidden layers (no batch normalization) on CIFAR-10 data.

Figure 19: Final accuracies and train times resulting from using the shrink and perturb trick with varying choices for $\lambda$ and noise scale. Missing numbers correspond to initializations that were too small to be trained. The bottom left entry is a pure random initialization while the top right is a pure warm start.

Figure 20: Complete learning curves corresponding to each entry of Figure 19, where $\lambda = 0$ is warm starting and $\lambda = 1$ is randomly initializing (plus the indicated noise amount).

### 8.2.4 Batch Online Learning Results for an MLP on SVHN (no batch normalization)

Here we show an online learning experiment, training an MLP consisting of three layers, ReLU activations, and 100-dimensional hidden layers (no batch normalization) on SVHN data.

**Test Accuracy**

| $\lambda$ \ Noise Scale | 1.0 | 0.1 | 0.01 | 0.001 | 0.0001 | 0.0 |
|---|---|---|---|---|---|---|
| 1.0 | 78.8 | 79.1 | 78.9 | 78.9 | 79.0 | 79.0 |
| 0.8 | 81.5 | 80.9 | 80.9 | 81.1 | 81.2 | 81.0 |
| 0.6 | 82.4 | 81.4 | 81.3 | 81.6 | 81.7 | 81.7 |
| 0.4 | 82.7 | 82.5 | 81.9 | 82.4 | 82.2 | 82.0 |
| 0.2 | 82.8 | 82.9 | | | | |
| 0.0 | 81.3 | 81.5 | | | | |

**Run Time (minutes)**

| $\lambda$ \ Noise Scale | 1.0 | 0.1 | 0.01 | 0.001 | 0.0001 | 0.0 |
|---|---|---|---|---|---|---|
| 1.0 | 61.2 | 37.3 | 37.6 | 35.8 | 36.1 | 37.6 |
| 0.8 | 93.8 | 84.8 | 82.0 | 82.2 | 79.2 | 81.8 |
| 0.6 | 133 | 115 | 127 | 119 | 125 | 127 |
| 0.4 | 188 | 159 | 160 | 157 | 166 | 173 |
| 0.2 | 264 | 211 | | | | |
| 0.0 | 361 | 398 | | | | |

Figure 21: Average final accuracies and train times resulting from using the shrink and perturb trick with varying choices for $\lambda$ and noise scale. We iteratively supply batches of 1,000 to the model and train it to convergence. Missing numbers correspond to initializations that were too small to be trained. The bottom left entry is a pure random initialization while the top right is a pure warm start.

Figure 22: Complete learning curves corresponding to each entry of Figure 21, where $\lambda = 0$ is warm starting and $\lambda = 1$ is randomly initializing (plus the indicated noise amount).

### 8.2.5 Batch Online Learning Results for a ResNet-18 on CIFAR-10 with weight decay

Here we show an online learning experiment, with a ResNet-18 on CIFAR-10 data, iteratively supplying batches of 1,000 to the model and training it to convergence with a weight decay penalty of .001. Note that this is aggressive regularization—increasing weight decay by an order of magnitude results in models that cannot reliably fit the training data.

| | Test Accuracy | | | | | | Run Time (minutes) | | | | | |
|---|---|---|---|---|---|---|---|---|---|---|---|---|
| $\lambda$ | 1.0 | 0.1 | 0.01 | 0.001 | 0.0001 | 0.0 | 1.0 | 0.1 | 0.01 | 0.001 | 0.0001 | 0.0 |
| 1.0 | 73.4 | 71.3 | 71.8 | 72.1 | 71.9 | 72.1 | 45.6 | 40.0 | 47.2 | 41.1 | 43.9 | 44.4 |
| 0.8 | 75.0 | 74.3 | 75.0 | 74.7 | 73.8 | 73.8 | 47.2 | 46.4 | 49.9 | 54.8 | 50.2 | 55.5 |
| 0.6 | 77.1 | 75.5 | 76.0 | 76.1 | 76.1 | 75.8 | 74.5 | 51.1 | 56.0 | 57.5 | 61.5 | 58.5 |
| 0.4 | 75.5 | 76.8 | 76.5 | 76.6 | 76.2 | 75.8 | 129 | 55.6 | 65.8 | 66.6 | 81.1 | 72.6 |
| 0.2 | 76.2 | 79.0 | | | | | 155 | 74.6 | | | | |
| 0.0 | 74.8 | 76.2 | | | | | 156 | 185 | | | | |

Figure 23: Average final performance and run times resulting from using the shrink and perturb trick with varying choices for $\lambda$ and noise scale. Missing numbers correspond to initializations that were too small to be trained. The bottom left entry is a pure random initialization while the top right is a pure warm start.

Figure 24: Complete learning curves corresponding to each entry of Figure 23, where $\lambda = 0$ is warm starting and $\lambda = 1$ is randomly initializing (plus the indicated noise amount).

### 8.2.6 Batch Online Learning Results for a ResNet-18 on SVHN with weight decay

Here we show an online learning experiment, with a ResNet-18 on SVHN data, iteratively supplying batches of 1,000 to the model and training it to convergence with a weight decay penalty of .001. Note that this is aggressive regularization—increasing weight decay by an order of magnitude results in models that cannot reliably fit the training data.

Figure 25: Average final accuracies and train times resulting from using the shrink and perturb trick with varying choices for $\lambda$ and noise scale. Here we show an online learning experiment with a ResNet-18 on SVHN data, iteratively supplying batches of 1,000 to the model and training it to convergence with a weight decay of .001. Missing numbers correspond to initializations that were too small to be trained. The bottom left entry is a pure random initialization while the top right is a pure warm start.

Figure 26: Complete learning curves corresponding to each entry of Figure 25, where $\lambda = 0$ is warm starting and $\lambda = 1$ is randomly initializing (plus the indicated noise amount).

### 8.2.7 Shrink and Perturb for Pre-Training

In this section we show the effect of applying the shrink and perturb trick at various noise scales in pre-training scenarios like those shown in Figure 9. In each experiment we pre-train a ResNet-18 on one dataset and then train to convergence on the the indicated fraction of a target dataset.

Figure 27: Shrink and perturb pre-train plots for various shrinkage parameters $\lambda$ and noise scale 1e-5.

Figure 28: Shrink and perturb pre-train plots for various shrinkage parameters $\lambda$ and noise scale 1e-4.

Figure 29: Shrink and perturb pre-train plots for various shrinkage parameters $\lambda$ and noise scale 1e-3.

Figure 30: Shrink and perturb pre-train plots for various shrinkage perameters $\lambda$ and noise scale 1e-2.

Figure 31: Shrink and perturb pre-train plots for various shrinkage perameters $\lambda$ and noise scale 1e-1.

## 9   Companion Figures

Figure 32: An online learning experiment, using CIFAR-10 data supplied to a ResNet in batches of 1000, using a learning rate schedule and SGD instead of a fixed learning rate with Adam.

Figure 33: A companion to Figure 11, showing validation accuracy as a function of different correlation measurements between warm-started model final weights and initializations.