[Reviews · NeurIPS 2020]

Review 1

Summary and Contributions: This paper addresses warm-starting as the problem of lack of performance when a model is retrained using an existing converged model as initialization. In particular, the paper focuses on those cases where the amount of data used for training the second model is larger than the first case. After a comprehensive analysis, the paper proposes to shrink and perturb as a way to improve retraining with new data.

Strengths: - Very relevant problem that does not seem to be discussed much in the literature. If solved, could have large impact for the community. - Simple method easy to implement and use in different scenarios

Weaknesses: - Experimental paper with little no theoretical contribution - Take home message seems not very convincing (see below). For instance, seems like this is particularly useful when the new model has limited data (which is far from the motivation in the introduction).

Correctness: - The paper is mostly based in empirical evidence and is conducted in a methodic manner. - There is a no clear take home message. It is not clear to me if, in the end, the original concern is solved: Is the proposed method able to reduce the training time of a network when additional data is added to the training set?

Clarity: The paper is clear and easy to read and follow.

Relation to Prior Work: This part seems correct to me.

Reproducibility: Yes

Additional Feedback: - Images do need some improvement. In the printed version, it is difficult to understand what is doing better or worse. For instance, in Figure 9, the proposed method performs at least as well as the best performing method. I think I can not see that from the figures.


Review 2

Summary and Contributions: The authors address the problem of training from "warm start" model and propose an approach to solve the problem. There are extensive empirical evaluations demonstrating that training from "warm start" model hurts the generalization. To solve this problem, the authors propose a shirk-perturb method that efficiently closes the generalization gap between training from "warm start" and a random model.

Strengths: 1. Extensive empirical evaluation. They empirically demonstrate the problem of worse generalization from learning warm-up model, as well as the effectiveness of the shirk-perturb method to solve this problem. 2. Worse generalization from warm start model is a common problem but few works addressed it formally. The authors empirically evaluate the problem by specific designed experiments.

Weaknesses: 1. I think the present work lacks sufficient analysis though there are many empirical validation results. It would be better if authors could reason more behind extensive empirical evaluations. 2. It's better to evaluate on a large-scale dataset like ImageNet.

Correctness: 1. A fixed small learning rate 0.001 could lead to a poor generalization, which affects the results of comparison. It might arrive a sharp local minimum when using small learning rate. For SGD a learning rate schedule should be involved. 2. In section 3.1, why choosing Pearson correlation to show the difference between optimized solution and its initialization?

Clarity: The paper is mostly well written. One suggestion is shrinking the font size of description sentences of figures.

Relation to Prior Work: I believe authors clearly present related works and their differences.

Reproducibility: Yes

Additional Feedback: As seen in the Figure 4, there is a place in first few epochs where it achieves both high test accuracy and less difference. It indicates that overfitting leads to a poor generalization when training from an over-fitted warm-start model. It would be interesting to investigate mode along the direction.


Review 3

Summary and Contributions: The authors of this article have made an extensive study of the phenomenon of overfitting when a neural network (NN) has been pre-trained: pre-training a neural network with 50% of available data, then training it with 100% of available data leads to poorer performance than training it directly with 100% of available data. They compare these two setups with different learning rates (LR), batch size, pre-training epochs, and regularization factor. Moreover, they demonstrate that altering the SGD update by shrinking and noising the updated weight prevents such overfitting.

Strengths: The authors have tested the main hyperparameters we usually tune when training a NN. Moreover, they have excluded from their study some tricks that might alter the fairness of their comparison, as LR schedule or data augmentation. Notably, they have proven experimentally that pre-training causes overfitting for a wide range of hyperparameters.

Weaknesses: The problem studied by the authors is not a major one. The authors do not explain their findings about the effect of pre-training, either experimentally or theoretically. It would be valuable to understand *why* such overfit is observed when training a pre-trained NN. I was more expecting an explanation than a solution to the problem. EDIT: after reading the other reviews and the rebuttal, I think there is a lack of either theoretical ground, or extensive experiments, that would have help to understand precisely this "warm-starting problem". The authors have run experiments with RNNs, which is helpful, but I still think that their observations should be validated in a wider range of tasks (e.g., regression...) and NN models (e.g., VGG or other CNNs). Even a negative result in one case would be interesting.

Correctness: The authors have tested enough sets of hyperparameters to validate their claim about the effect of pre-training (LR, weight decay, number of pre-training epochs, batch size). The proposed technique ("shrink and perturb", Section 4) seems to be tested only with ResNet-like architectures.

Clarity: There is no major writing issue.

Relation to Prior Work: The authors have cited main papers about weight initialization, and also a paper about the link between overfitting and distance of the weights from their initialization. This last one corroborates the experimental results of the paper. Apparently, the studied problem has not been addressed before.

Reproducibility: No

Additional Feedback:


Review 4

Summary and Contributions: The paper addresses the problem of coming with tricks to warm-start network. The authors demonstrate that warm-starting / fine-tuning from existing weights causes drop in generalization performance. To mitigate this problem they propose the use of "shrink and perturb", which scales the weights and adds noise to the weights.

Strengths: The paper brings light upon the phenomena where training from scratch performs better than warm-starting / fine-tuning from on CIFAR10. Although this does not seems to be a previously noted phenomena, if this behavior is actually a commonly occurring phenomenon in continual learning this could be of value to the community and open up future research directions. The paper shows that this phenomena is robust to regularization, couple of models .. and provides various empirical studies. The paper provides a method called "shrink and perturb" that allows one to train models with warm-starts.

Weaknesses: The paper is limited to evaluating on CIFAR/SVHN, and I worry that this phenomenon may not extend to other methods and tasks. Warm-starting .. in the context of the problem setup of the authors .. seems to be basically the same thing as fine-tuning with more-data. This phenomenon doesn't seem to be happening on more sophisticated computer-vision tasks, and finetuning from datasets like ImageNet leads to similar or better performance with much faster convergence. Although the label-space is different in many fine-tuning setups one can imagine extending the existing setup to cover common and more realistic problems. The paper is written to motivate the idea of re-using weights on for continual/online learning setting but splitting the datasets into 2 sets (training with 1 and fine-tuning with both) seems to me a little toyish and unconventional continual learning setting. In online / continual learning there is a distribution shift as the dataset enters, but the dataset seems to be randomly split meaning that on expectation the distribution of these 2 sets should be the same. It would have been interesting to see the current setup on a distribution shifting datasets? Furthermore, due to the size of the dataset used in the experimentts, i wonder if the model is just overfitting to the training set (50% dataset) .. which makes it harder for the model to recover from. What happens if you don't wait till 100% convergence? There seems to be not enough theory behind why the behavior of generalization drop happens. Although I am not against the empirical findings on phenomena, the motivation for why shrink and perturb solves this seems a little lack luster. Either having a stronger theory, or demonstrating empirical result on a wide-range of realistic and challenging tasks would have made this paper stronger.

Correctness: Yes, the methods are clear.

Clarity: The paper is well written and easy to read.

Relation to Prior Work: To my knowledge, the paper proposes to solve a problem that has not been tackled by prior works.

Reproducibility: Yes

Additional Feedback:

[Author Response · NeurIPS 2020]

1. First, we would like to thank the reviewers for their time and effort.

2. **Reviewer 1:** From your brief review, we cannot tell why you believe there to be no take-home message. We describe the warm-start problem, which you note is both important and understudied. We then describe a simple solution that remedies the issue. The original concern *is* resolved by the shrink and perturb trick.

Settings described in the introduction are just batch online learning scenarios, which we faithfully simulate in our main experiments (e.g. Figure 7). Please see response 3 under reviewer 4. Final-version plots will be made printer friendly.

**Reviewer 2: 1.** Please see Section 3.1 for a detailed analysis of both learning rate and batch size. Figure 3 shows that, while there exist some hyperparameter values for which warm-started models perform as well as randomly-initialized models, they take as long to train as randomly-initialized models. Tuning these parameters can close the genearlization gap, but only by sacrificing the computational efficiency we would like to see from warm starting.

Separately, it is true that it is commonplace to train models using SGD with a learning rate schedule. To address this concern, we performed an online learning experiment where, starting from 100 CIFAR-10 samples, batches of 10000 points are supplied to the learner in sequence. Using a fresh, warm, or shrink-perturb initialization, we train the ResNet using SGD for 350 epochs using the standard learning rate schedule. As expected, a generalization gap between warm-started and randomly-initialized models still exists and shrink-perturb is able to remedy the problem (Fig A). Similar to warm-start plots in the Appendix, the final copy will include a detailed analysis of scheduled optimization. Models were all trained for the same number of epochs, but randomly-initialized models take much longer to converge.

**2.** We use Pearson to study the similarity between warm-start and final-solution weights. Other notions of correlation, like cosine similarity, Spearman, and euclidean distance provide similar plots. We will show these in the final version.

**3.** It may be difficult to tell from Figure 4, but these models that have not been trained much actually perform significantly worse than fully-trained models. This might be more clear in Table 3, where we apply aggressive regularization to prevent overfitting and still observe a generalization gap. We will make Figure 4 more clear in light of this discussion.

**Reviewer 3: 1.** In the Appendix we have extensive experiments using shrink perturb with multilayer perceptrons, rather than only ResNets. We also experiment with batch normalization and weight decay, and will add RNNs (see **1** below).

**Reviewer 4: 1.** Indeed, this article focused on experiments with image datasets using ConvNets and MLPs. In response to this note, we performed an online learning experiment using a very different architecture and dataset: a two-layer, bidirectional RNN on the IMDB movie reviews dataset, where the task is to predict whether a movie review is positive or negative. We iteratively supply batches of 500 samples to the model, and compare randomly-initialized, warm-started, and shrink-perturb models in the figure below. We show that there exists a performance gap between randomly-initialized and warm-started models, which the shrink and perturb trick closes (Fig B). The analysis here is truncated due to time constraints, but the trend is clear; we will include a thorough analysis in the final copy.

**2.** It is not true that this phenomenon does not happen in vision literature. For example, active learning work in image classification requires models to be trained from scratch at each round [7, 8]. Not all articles say this explicitly, but popular deep active learning github repositories also show that random reinitialization after selection is necessary [9,10]. Fine tuning on ImageNet works because the source dataset is large and the target datasets are, comparatively, significantly smaller. Please see Section 4.2 which discusses the pre-training setup in detail. We show that when the target dataset is large, it is often better to initialize models from scratch than it is to pre-train on a source dataset. Figure 9 shows that shrink-perturb is effective in this setting as well: models initialized with shrink-perturb perform as well or better than either randomly initializing or pre-training, regardless of whether pre-training is helpful.

**3.** We believe there might be some confusion here. While we do "toy" two-phased experiments for illustrative purposes in Table 2.1 and Figures 3-5, our primary results are for full online learning scenarios. These experiments are like Figure 7, where samples are iteratively provided to the learner in batches of 1000. Each time the model receives new data, it is appended to the current training set and is trained to convergence before receiving the next batch. These experiments are consistent with what is usually done in active and online learning work. Also note that large tabular figures, like figure 8, also show this fully-online setting. Here, to concisely report results, we are showing only the final accuracies and train times, i.e., after dozens of rounds of online learning. That is, the plots in Figure 7 correspond to a single column of Figure 8. This is likewise true for all of the similar appendix figures. We will make this more clear in the final version.

**4.** The warm-start problem is not induced by overfitting. Figure 4 shows that if we do not train to convergence, we still observe a generalization gap. As mentioned above, Table 3 also shows that when overfitting is avoided, by applying regularization, a generalization gap is still present. Also note that the shrink-perturb trick is able to efficiently resolve the generalization gap present in regularized models as well, as shown in Appendix 6.2.5 and 6.2.6.



[Meta-Review · NeurIPS 2020]

The paper reports an interesting phenomenon -- sometimes fine-tuning a pre-trained network does worse than training from scratch, even when pre-training and fine-tuning are performed on the same dataset. The authors propose a method to remedy this problem. The reviewers are on the fence about the paper, but acknowledge that's its an understudied area. Their main concern is lack of any theoretical insights and the method being a "trick". I believe that findings of this paper are going to be of interest to the community. I recommend the authors to investigate if there are any theoretical insights to be gleaned from the reported empirical results.